# Methylphenidate Analogues as a New Class of Potential Disease-Modifying Agents for Parkinson’s Disease: Evidence from Cell Models and Alpha-Synuclein Transgenic Mice

**DOI:** 10.3390/pharmaceutics14081595

**Published:** 2022-07-30

**Authors:** Andrea Casiraghi, Francesca Longhena, Gaia Faustini, Giovanni Ribaudo, Lorenzo Suigo, Gisela Andrea Camacho-Hernandez, Federica Bono, Viviana Brembati, Amy Hauck Newman, Alessandra Gianoncelli, Valentina Straniero, Arianna Bellucci, Ermanno Valoti

**Affiliations:** 1Department of Pharmaceutical Sciences, University of Milan, Via Luigi Mangiagalli 25, 20133 Milano, Italy; a_casiraghi@outlook.com (A.C.); lorenzo.suigo@unimi.it (L.S.); ermanno.valoti@unimi.it (E.V.); 2Department of Molecular and Translational Medicine, University of Brescia, Viale Europa 11, 25123 Brescia, Italy; francesca.longhena@unibs.it (F.L.); gaia.faustini@unibs.it (G.F.); giovanni.ribaudo@unibs.it (G.R.); federica.bono@unibs.it (F.B.); v.brembati@unibs.it (V.B.); alessandra.gianoncelli@unibs.it (A.G.); arianna.bellucci@unibs.it (A.B.); 3Medicinal Chemistry Section, Molecular Targets and Medications Discovery Branch, NIDA-IRP, 333 Cassell Drive, Baltimore, MD 21224, USA; giselaandrea.camachohernandez@nih.gov (G.A.C.-H.); anewman@intra.nida.nih.gov (A.H.N.)

**Keywords:** Parkinson’s disease, α-synuclein/synapsin III complex, methylphenidate analogues, *threo* methyl 2-(piperidin-2-yl)-2-(p-tolyl)acetate hydrochloride, motor recovery effect

## Abstract

Parkinson’s disease (PD) is characterized by dopaminergic nigrostriatal neurons degeneration and Lewy body pathology, mainly composed of α-synuclein (αSyn) fibrillary aggregates. We recently described that the neuronal phosphoprotein Synapsin III (Syn III) participates in αSyn pathology in PD brains and is a permissive factor for αSyn aggregation. Moreover, we reported that the gene silencing of Syn III in a human αSyn transgenic (tg) mouse model of PD at a pathological stage, manifesting marked insoluble αSyn deposits and dopaminergic striatal synaptic dysfunction, could reduce αSyn aggregates, restore synaptic functions and motor activities and exert neuroprotective effects. Interestingly, we also described that the monoamine reuptake inhibitor methylphenidate (MPH) can recover the motor activity of human αSyn tg mice through a dopamine (DA) transporter-independent mechanism, which relies on the re-establishment of the functional interaction between Syn III and α-helical αSyn. These findings support that the pathological αSyn/Syn III interaction may constitute a therapeutic target for PD. Here, we studied MPH and some of its analogues as modulators of the pathological αSyn/Syn III interaction. We identified 4-methyl derivative **I-*threo*** as a lead candidate modulating αSyn/Syn III interaction and having the ability to reduce αSyn aggregation in vitro and to restore the motility of αSyn tg mice in vivo more efficiently than MPH. Our results support that MPH derivatives may represent a novel class of αSyn clearing agents for PD therapy.

## 1. Introduction

The neuronal phosphoprotein Synapsin III (Syn III) plays a key role in the control of dopaminergic striatal neurotransmission [1]. We recently described that the functional interaction between Syn III and α-synuclein (αSyn), a synaptic enriched protein involved in the pathophysiology of several neurodegenerative disorders including PD and Lewy body (LB) dementia (DLB) [2], is crucial to ensure dopamine (DA) neurotransmission [3]. In the brain of patients affected by PD and DLB, the pathological deposition of aggregated αSyn is thought to initiate neurodegeneration [2,4,5,6]. In particular, multiple pieces of evidence support that αSyn micro-aggregation at synaptic terminals can trigger a retrograde synapse-to-cell body degeneration process culminating in neuronal cell death [4,7,8,9,10]. Of note, we found that in the brain of patients affected by sporadic PD, Syn III participates in LB pathology and composes αSyn insoluble fibrils [11]. We then observed that Syn III controls αSyn aggregation, as its absence in knock-out (ko) mice prevented the formation of fibrillary αSyn deposits as well as the related degeneration of nigrostriatal neurons in an in vivo adeno-associated viral vector (AAV)-based model of PD [12]. Still, in an interventional experimental design, we observed that by inducing the gene silencing of Syn III in a human αSyn transgenic (tg) mouse model at PD-like stage characterized by advanced αSyn aggregation and overt striatal DA release failure, we could reduce αSyn insoluble fibrillary deposits and dopaminergic dysfunction. Synaptic vesicles clumping, striatal dopaminergic fibers loss and motor function impairment were also recovered [13]. This supports that Syn III consolidates αSyn aggregates, while its downregulation enables their reduction and redeems the PD-like phenotype.

Intrigued by findings showing that Syn III gene (SYN III) polymorphisms impact the therapeutic response to the monoamine reuptake inhibitor *threo*-methylphenidate (MPH) in attention deficit hyperactivity disorder (ADHD) patients [14], and that the effect of MPH on DA overflow and compartmentalization are mediated by αSyn [15], we investigated whether the αSyn/Syn III interplay could be modulated by MPH. We found that the pathological αSyn/Syn III co-deposition boosts the locomotor response to MPH [16]. Interestingly, this effect is not mediated by the DA transporter (DAT)-inhibitory action of MPH, as it is not perturbed by the administration of the selective DAT blocker GBR−12935, but rather depends on the presence of Syn III [16], suggesting a direct binding between Syn III and MPH. In support of this hypothesis, we developed a cell permeable fluorescent MPH analogue and confirmed its direct interaction with Syn III by acceptor photobleaching fluorescence resonance energy transfer (FRET) [17]. Moreover, by using *in silico* studies, we found that MPH stabilizes αSyn in a functional and low aggregation prone α-helical conformation which exhibits improved ability to interact with Syn III [16].

Collectively, these findings support that the αSyn/Syn III complex can constitute a druggable therapeutic target for PD, and that MPH therapeutic efficacy in patients in advanced stages of disease with freezing of gait [18,19], which should not respond to MPH treatment in light of their marked reduction in striatal DAT [20], may be ascribed to the ability of this drug to stimulate the functional αSyn/Syn III interaction.

In particular, we hypothesized that MPH could promote the establishment of an operative complex between Syn III and functional α-helical αSyn. The establishment of this αSyn/Syn III interaction will then promote the recovery of DA release and consequently of motor abilities [3]. In addition, by exerting this function, MPH may reduce the pathological interaction between Syn III and aggregated αSyn, resulting in fibril destabilization in a manner similar to Syn III gene silencing.

On this basis, we chose some known MPH analogues that may exhibit an improved ability to promote the functional αSyn/Syn III complex [21], when compared to MPH.

We started evaluating compound **I** in Figure 1, which is characterized by a small alkyl electron-donating group in *para* position of the phenyl moiety of MPH. We prepared both isomeric forms, *threo* and *erythro*, to verify if the functional αSyn/Syn III complex may have some structural requirement.

We then moved on to consider only the *threo* isomers, and we focused on differently hindered aromatic groups (compounds **II-*threo*** and **III*-threo*** in Figure 1) or on amine homologation (compound **IV-*threo***).

We tested the compounds for their ability to promote αSyn/Syn III interaction and their toxic potential by acceptor photobleaching FRET and 3-(4,5-dimethylthiazol-2-yl)-2,5-diphenyltetrazolium bromide (MTT) assay, respectively. Then, we assessed the ability of MPH and of the compounds, displaying an improved ability to promote αSyn/Syn III interaction, to reduce αSyn pathological aggregates in a cell model of PD. These studies led to the recognition of MPH analogue **I-*threo*** as the lead compound. Therefore, we also assessed whether acute i.p. treatment with 10 mg/kg of **I-*threo*** could stimulate the locomotor activity of aged (12-month-old) human αSyn tg mice. Notably, we found that this compound could stimulate the motility of these animals more efficiently than MPH. Although the ability of **I-*threo*** to interact with both the DAT and norepinephrine transporter (NET) [22] is documented, this action should not significantly impact the motor recovery effect of this molecule as the human tg mouse model used for this study shows a significant reduction in functional DAT at presynaptic membranes as this is retained within αSyn pathological synaptic deposits [23], as occurs in the brain of PD patients [24]. Our findings showing that the motility recovery induced by the acute administration of MPH in human αSyn tg mice is abolished by Syn III gene silencing [16] further support that the motor-stimulating effect of compound **I-*threo*** also depends on its capability to modulate Syn III.

Additionally, for the first time, we rationalized the binding of MPH and its analogues to the αSyn/Syn III complex using computational tools. These protein–protein docking studies provided a putative structure of the αSyn/Syn III assembly. This complex was analyzed to highlight the presence of novel ligand binding sites. This 3D macromolecular structure provided the opportunity to perform molecular docking to investigate the interaction pattern of drug candidates, in combination with molecular dynamics simulations.

## 2. Materials and Methods

### 2.1. Chemistry

#### 2.1.1. General

Starting materials and solvents were purchased from commercial suppliers and were used without further purification. Reaction conditions and yields were not optimized. ^1^H and ^13^C NMR spectra were acquired on a Varian 300 Mercury NMR spectrometer operating at 300 MHz for ^1^H NMR, and 75 MHz for ^13^C NMR; the chemical shifts are reported in ppm. Signal multiplicity is used according to the following abbreviations: s = singlet, d = doublet, dd = doublet of doublets, t = triplet, td = triplet of doublets, q = quadruplet, m = multiplet, sept = septuplet, and bs = broad singlet. Melting points were measured on a TA Instruments Q20 DSC system. TLC were performed on standard analytical silica gel layers (thickness 200 µm; aluminum support silica gel 60 matrix with fluorescent indicator 254 nm, Sigma-Aldrich/ Merck KGaA, Darmstadt, Germany). Chromatographic purifications were performed, in normal phase, using flash chromatography on Sepachrom Puriflash XS 420, and over different Biotage SNAP Ultra flash chromatography cartridges, filled with Merck Silica Gel 60 (0.040–0.063 μm). The purity of final compounds was assessed on HPLC by using Elite LaChrom HPLC system with diode array detector (190–400 nm) and a Water XBridgeTM C−18 column (5 µm, 4.6 × 150 mm). All the compounds are >95% pure by HPLC.

The specific synthetic protocols, together with ^1^H and ^13^C NMR, as well as characterization data, HPLC chromatograms and DSC analysis are reported in the Appendix A.

#### 2.1.2. Synthesis

The selected MPH analogues **I-*threo* [25,26]**, **I-*erythro* [26]**, **II-*threo* [26,27,28]**, **III-*threo* [26]** and **IV-*threo* [27,29]** have been previously described as psychostimulants and were synthetized following different synthetic pathways.

We developed a synthetic scheme for the synthesis of compounds **I-*threo***, **I-*erythro***, **II-*threo***, **III-*threo***, **IV-*threo*** based on the one proposed by Thai and coworkers [30] for other MPH analogues (Figure 1). We included some modifications in terms of intermediates and reagents, in order to maximize conversions and optimize reaction conditions.

Commercially available racemic pipecolic acid was protected at the piperidine nitrogen with Boc (**1**), prior to conversion into the corresponding morpholine amide **2** using TBTU (*O*-(benzotriazol−1-yl)-*N*,*N*,*N′*,*N′*-tetramethyluronium tetrafluoroborate) as a coupling agent. Where Thai and collaborators used the Weinreb amide, we used the morpholine amide, with higher yields, easier amide purification and the use of cheaper reagents. Reaction with a suitably substituted aryl lithium salt, generated in situ by the treatment of the corresponding aryl bromides with *n*-BuLi, afforded aryl ketones **3a**–**c**. Treatment of **3a**–**c** with methyltriphenylphosphonium bromide gave vinyl derivatives **4a**–**c**, which were then converted into the respective primary alcohols **5a**–**c** via hydroboration.

At this stage, the *threo* and *erythro* racemates, resulting from the generation of the second stereocenter, were readily separated by column chromatography on silica gel, and then **6a**–**c** *threo* and **6-a** *erythro* were oxidized to the corresponding carboxylic acids. The reaction with trimethylsilyldiazomethane (TMSCHN_2_) afforded methyl esters **7a**–**c**, which were not isolated and directly deprotected on the amine function under acidic conditions, yielding compounds **I-*threo***, **I-*erythro***, **II-*threo*** and **III-*threo***. The treatment of **I-*threo*** with formic acid and formaldehyde under reductive amination conditions yielded compound **IV-*threo***.

### 2.2. Animals

Twelve-month-old C57 BL/6 J wild-type (wt) and αSyn tg mice expressing the human C-terminally truncated 1–120 form of αSyn in the nigrostriatal system on an αSyn null background (SYN120 tg) [16,31,32] were used for this study. Mice were bred in our animal house facility at the Department of Molecular and Translational Medicine of University of Brescia, Brescia, Italy. Animals were maintained under a 12 h light–dark cycle, at 22 °C, and had ad libitum food and water. All experiments were carried out in accordance with Directive 2010/63/EU of the European Parliament and of the Council of 22 September 2010 on the protection of animals used. All experimental procedures conformed to the National Research Guide for the Care and Use of Laboratory Animals and were approved by the Animal Research Committees of the University of Brescia (Protocol Permit 719/2015-PR and 261/2020-PR). All achievements were made to minimize animal suffering and to reduce the number of mice used.

### 2.3. Acceptor Photobleaching FRET Studies

Acceptor photobleaching FRET detects the proximity of fluorescently labelled molecules over distances >100 Å [33]. The phenomenon involves the non-radiative transfer of energy from a donor fluorophore in its excited state to a nearby acceptor. FRET can occur between a pair of molecules that approximately need the same amount of energy to transition from their ground state to excited states. In particular, the energy released when an excited donor transitions to the ground state must correspond to the energy required to excite the acceptor. The couples of fluorescent molecules presenting the typical characteristics of a donor and an acceptor are defined as FRET pairs [34]. When the donor fluorophore is excited, it is able to transfer its excitation energy to a neighboring acceptor fluorophore [35]. The energy transfer efficiency is related to the distance separating a given donor and acceptor pair. As the proximity between the fluorophores is dependent on the interaction of the linked proteins, the FRET efficiency is strictly related to this parameter. In the case of acceptor photobleaching FRET, the efficiency is measured as the recovery of the donor after the bleaching of the acceptor within a FRET couple. In our experimental setup, the FRET couple was represented by green fluorescent protein (GFP)-tagged human 63 kDa Syn III isoform (GFP-Syn III), which is the most abundant in the adult brain [36], and red fluorescent protein (RFP)-tagged human wild-type αSyn (RFP-αSyn). Human neuroblastoma SK-N-SH cells were grown in complete medium comprising Dulbecco’s modified Eagle medium with 1000 mg glucose/L supplemented with 10% heat-inactivated fetal bovine serum, 100 μg/mL penicillin and 100 μg/mL streptomycin. Cells were maintained at 37 °C under a humidified atmosphere of 5% CO_2_ and 95% O_2_. For FRET studies, SK-N-SH cells were prepared as previously described [16]. Briefly, cells were seeded onto poly-D-lysine-coated 13 mm glass coverslips in 24-well plates (15,000 cells per coverslip) and were maintained in differentiation medium for ten days, by adding 10 μM retinoic acid daily to the medium as previously described [16,37]. The plasmids encoding GFP-Syn III and RFP-αSyn were transiently transfected in coverslip-plated SK-N-SH neuroblastoma cells that had previously undergone 10 days of neuronal-like differentiation, as mentioned above. Syn III was inserted in the pEGFP-C3 plasmid, thanks to XhoI and XbaI cloning sites, whereas αSyn was inserted in the pCMV6-AN-mRFP plasmid, thanks to SfgI and Hind III cloning sites. In both cases, the fluorophore is located at the N-terminus of the protein of interest. Three days after transfection, cells were treated for 15 min with vehicle (normal saline 0.9%—Basal) or 10 μM MPH (d-*threo*) or 10 μM MPH analogues (only for **I-*threo*** we also probed the effect of 0.05 and 0.1 µM) and then immediately fixed with 4% paraformaldehyde (PFA, Bio-Optica) for 15 min. The fixed coverslip-plated SK-N-SH cells were subsequently mounted on glass slides and used for acceptor photobleaching FRET experiments. These were carried out by using a Zeiss LSM 880 confocal laser microscope (Carl Zeiss) with the laser set on λ = 488–543. Following the identification of RFP-αSyn- and GFP-Syn III-positive cells, three-to-six regions of interest (ROI) were analyzed for 4 series of images. Before each FRET acquisition, two images of basal condition (pre-bleaching) were acquired before bleaching the RFP acceptor fluorophore with the 543 laser set at 65% power. Two other images were then acquired after the bleaching (post-bleaching). FRET efficiency (corresponding to the GFP-Syn III recovery after RFP-αSyn photobleaching) was measured by using Zen Black 2.3 software, Carl Zeiss, Germany. The average intensity of the image background (outside of the positive cells) was subtracted from the average intensity of the ROI. All the FRET values resulting from the different ROIs were used for statistical analysis.

### 2.4. Live Fluorescence Lifetime Imaging (FLIM)

FRET-based FLIM is a method that measures fluorescence lifetime as the average time that the donor spends in the excited state after the absorption of light, prior to returning to the ground state by emitting a photon. The method can be applied to live cells allowing live imaging. Donor fluorescence lifetime is decreased when FRET occurs between the fluorophore couple, in our case the GFP-RFP pair [38]. For live cell imaging of FRET-based FLIM analysis, the SK-N-SH cells were seeded onto poly-D-lysine-coat 1.7 cm^2^ glass chamber (15,000 cells per well) and were maintained in differentiation medium, for ten days, by adding 10 μM retinoic acid to the medium daily. At day 7, cells were transiently transfected with the plasmids transducing the expression of GFP-Syn III and RFP-αSyn. Cells transfected with only GFP-Syn III were used as negative controls. Three days after transfection, cells were placed under a confocal Zeiss LSM 880 microscope, operated with ZEN software (Zeiss), equipped with an environmental chamber that maintains the temperature at 37 °C and with a supply of humidified 5% CO_2_ to the cells. Cells were imaged with 488 and 594 nm confocal lasers for GFP and RFP, respectively, with a 63× oil immersion objective, immediately prior to FLIM of the same field of view. Cells were treated with vehicle (normal saline 0.9%—Basal), 10 μM MPH (d-threo), 10 μM **I-*threo*** or 10 μM **I-*erythro***, and the images were recorded 15 min after the treatment. The time-domain FLIM was measured using a Time-Correlated Single Photon Counting (TCSPC) (PicoQuant) operated with SymPhoTime 64 software (PicoQuant). The two-photon laser (Chameleon), integrated with a Zeiss 880 microscope and operated with ZEN software, generated the required pulsed illumination with a repetition rate of 80 MHz and pulse width of 500 fs. For GFP excitation, the wavelength was set to 860 nm. The images of GFP fluorescence generated in ZEN software were stored and processed with SymPhoTime 64 2.7, PicoQuant, Berlin, Germany. The mean fluorescence intensities within cell body were determined using the ROI tool in SymPhoTime 64. The mean intensity of the background measured in several regions that did not contain cells was further subtracted from the fluorescence intensity data. The fluorescence lifetime analysis was performed in the SymPhoTime 64 software within the cell body, automatically recognized by the software as ROI 0. Individual photon arrivals were detected using a SPAD detector, and events were recorded by a PicoHarp 300 TCSPC module. All measured photons of ROI 0 are combined into a global histogram, representing the decay curve used to obtain the fitting parameters. Mono- and bi- exponential fittings were obtained for GFP alone and in the presence of RFP, respectively. Parameters obtained from SymphoTIme software include Lifetime (τ) and Amplitude (A). For ROI 0, the amplitude-weighted average lifetime of GFP is calculated using the following equation [39]:$$\tau_{\mathrm{AvAmp}}=\frac{\sum_{k=0}^{n-1}A\left[k\right]\tau \left[k\right]}{A_{Sum}}$$

The samples of GFP alone were performed under identical conditions in the absence of RFP. The FLIM-FRET percentage efficiency was calculated as follows: 100 × [1 − (lifetime of donor with FRET/lifetime of donor without FRET)]. For lifetime analysis, fifteen cells were recorded for each condition, included GFP alone.

### 2.5. Treatments with MPH-Analogues in the SK-N-SH Cells Overexpressing Human Full Length αSyn 

For **I-*threo*** and **IV-*threo*** treatment, SK-N-SH neuroblastoma cell line stably overexpressing αSyn were used as an in vitro model of synucleinopathy as they exhibit αSyn aggregation, as it has been previously reported that three days αSyn overexpression is sufficient to induce protein aggregation [40]. Briefly, cells were seeded onto poly-D-lysine-coated glass cover slides in 24-well plates (10,000 cells/well) and maintained at 37 °C under a humidified atmosphere of 5% CO_2_ and 95% O_2_ in Dulbecco’s modified Eagle medium with 1000 mg glucose/L (Sigma-Aldrich/ Merck KGaA, Darmstadt, Germany), 10% heat-inactivated fetal bovine serum, 100 μg/mL penicillin, 100 μg/mL streptomycin and 0.01 μM non-essential amino acids (Sigma-Aldrich/ Merck KGaA, Darmstadt, Germany). When the cells reached 80% confluence, these were treated with **I-*threo*** 0.05, 0.1, 10 μM or with **IV-*threo*** 10 μM for 24 h, to assess the effect of the compounds on αSyn aggregation. Cells were then fixed with 4% PFA (Bio-Optica, Milan, Italy) for 15 min, at 4 °C (n = 3 replicates).

### 2.6. I-threo and IV-threo Treatments in Primary Mesencephalic Neurons Subjected to Glucose Deprivation (GD)-Induced αSyn Aggregation

Primary ventral mesencephalic cells from C57 BL/6 J 13.5 days embryos were isolated and cultured as previously described [3]. Briefly, after 10 min enzymatic dissociation, the single cell suspension was diluted in Neurobasal medium (Life Technologies, Carlsbad, CA, USA) added with 100 μg/mL penicillin, 100 μg/mL streptomycin Sigma-Aldrich/ Merck KGaA, Darmstadt, Germany), 2 mM glutamine Sigma-Aldrich/ Merck KGaA, Darmstadt, Germany) and 2% B27 supplement (Life Technologies, Carlsbad, CA, USA). Cells were then centrifuged, counted, and seeded onto poly-D-lysine-coated glass cover slides in 24-well plates for immunocytochemistry (80,000 cells/well). Seeded cells were maintained at 37 °C under a humidified atmosphere of 5% CO_2_ and 95% O_2_ in Neurobasal medium (Life Technologies, Carlsbad, CA, USA), and after 7 days in vitro (DIV), they were exposed to GD in order to induce αSyn aggregation [3]. GD was performed by incubating the cells with Phosphate Buffer (Sigma-Aldrich/ Merck KGaA, Darmstadt, Germany) supplemented with 2 mM glutamine and 2% of B27 for 1 h, at 37 °C. After GD, the cells were treated with different concentrations of **I-*threo*** and **IV-*threo*** for 24 h and fixed in 4% PFA (Bio-Optica, Milan, Italy) for 15 min, at 4 °C (n = 3 replicates).

### 2.7. MTT Assay

The cytotoxic effect of compounds treatment was measured by evaluating mitochondrial dehydrogenase activity using the MTT salt assay. Primary neuronal cultures from mesencephalic tissues were dissected from C57 BL/6 J 13.5-day-old embryos. Briefly, after mechanical dissociation, the single cells were re-suspended in Neurobasal medium (Life Technologies, Carlsbad, CA, USA) containing 100 μg/mL penicillin, 100 μg/mL streptomycin, 0.5 mM glutamine and 1% B27 supplement. Cells were then centrifuged and seeded onto poly-D-lysine-coated 24-well plates (100,000 cells/well). Cells were maintained at 37 °C under a humidified atmosphere of 5% CO_2_ and 95% O_2_ for at least 10 days in vitro prior to their use to allow their maturation. At DIV 10, cells were treated with increasing concentrations (0.01 µM, 0.1 µM, 1 µM, 10 µM, and 100 µM) of the different MPH analogues. Twenty-four hours after the treatments, culture medium was removed, and 300 μL of MTT 0.5 mg/mL diluted in culture medium was added to each well. After 90′ incubation at 37 °C, the medium was removed, and 250 μL aliquots of DMSO were added to each well to solubilize the formazan crystals. Absorbance was measured at 570 nm using a microplate reader. Cell viability was expressed as absorbance value (n = 3 replicates).

### 2.8. Immunocytochemistry

After 24 h of treatment with the MPH analogues, SK-N-SH cells and primary neurons were fixed by incubating for 15 min in 4% PFA (Bio-Optica), at 4 °C. Coverslips were incubated for 1 h, at room temperature, in blocking solution (2% *w*/*v* bovine serum albumin (BSA, Sigma-Aldrich/ Merck KGaA, Darmstadt, Germany) plus 3% *v*/*v* normal goat serum (Sigma-Aldrich/ Merck KGaA, Darmstadt, Germany) in PBS, then overnight, at 4 °C, with αSyn antibody (1:600, BD, Cat. No. 610787) in the case of SK-N-SH cells, and with Syn III antibody (1:500, Synaptic System, Cat. No. 106 303) in the case of mesencephalic neurons. On the following day, SK-N-SH cells overexpressing αSyn were incubated for 1 h, at room temperature, with the Cy3 fluorochrome-conjugated secondary antibody (Jackson Immunoresearch, West Grove, PA, USA) diluted in 0.1% Triton X−100 PBS plus BSA 1 mg/mL. The following day, mesencephalic neurons were incubated for 1 h, at room temperature, with the Alexa 488 fluorochrome-conjugated secondary antibody (Jackson Immunoresearch, West Grove, PA, USA) diluted in 0.1% Triton X−100 PBS plus BSA 1 mg/mL. After that, primary neurons were incubated with the second primary antibody for αSyn antibody (1:600, BD, Cat. No. 610787) for 2 h at room temperature and then with Cy3 fluorochrome-conjugated secondary antibody (Jackson Immunoresearch, West Grove, PA, USA), diluted as mentioned above. After three washes with 0.1% Triton X−100 PBS, cell nuclei were counterstained with ToPro^TM^−3 dye (Life Technologies, Carlsbad, CA, USA), and the coverslips were mounted onto glass slides using Vectashield (Vector Laboratories, Burlingame, CA, USA).

### 2.9. Open Field Beam Walking Behavioral Tests

Acute locomotor activity of twelve-month-old SYN120 tg treated with vehicle or MPH or **I-*threo*** 10 mg/kg was analyzed with the automated AnyMaze video tracking system (Stoelting), as described before. Total distance travelled was recorded in the open field arena (50 × 50 cm^2^), data were analyzed in 14 trials of 2.5 min each. First, mice were placed in the arena to explore the environment for 5 min before starting the test. After that, 10 min of registration in basal condition were recorded, and then mice were injected with MPH or **I-*threo*** (intra peritoneal (i.p.) 10 mg/kg, dissolved in normal saline 0.9%), while control mice were treated with vehicle (normal saline 0.9%), placed again in the open field arena, and recorded for the following 25 min. All the experiments were conducted during daylight hours. n = 3–4 animals per group were analyzed.

Time spent to traverse the beam in the beam walking test was assessed as previously described by our group [13]. Briefly, the beam (2.5 cm width and 1 m length) was placed at 25 cm height supported by lined-up cages, ending into the animal’s home cage. The day before the experiment, beam walking training was performed four times every 30 min for each animal: the mouse was allowed to traverse the entire length by placing it at the beginning of the beam and keeping the home cage in close proximity to the animal to encourage it to walk along the beam. On the test day, the time taken to cross the beam was measured and evaluated starting from when the mouse began to move forward and ending when the first forepaw was placed outside the beam.

### 2.10. Confocal Microscopy

The acquisition of images from primary neuronal cells was performed by using a LSM 880 Zeiss confocal laser equipped with Airyscan detector (Carl Zeiss) with λ = 405–488–543 nm lasers. In particular, z-stack images were acquired with the height of the sections scanning = 1 μm. Images (512 × 512 pixels) were processed by Airyscan processing and maximum intensity projection by using Zen Black 2.3 Software, Carl Zeiss, Germany.

### 2.11. Statistical Analysis

Differences between the FRET efficiency/Amplitude (FLIM) of GFP-Syn III and RFP-αSyn overexpressing SK-N-SH cells exposed to MPH or MPH-analogues were assessed by one-way ANOVA followed by Tukey multiple comparison test. At least n = 15 cells for each experimental condition were analyzed. In each cell, 3–6 different ROIs were acquired, and the resulting median value for each cell was then plotted and subjected to statistical analysis. Differences between αSyn-positive particle average size treated with the compounds were assessed by one-way ANOVA followed by Newman–Keuls multiple comparison test. At least n = 15 images for each experimental condition were analyzed, and the resulting quantification for each image was then plotted and subjected to statistical analysis. Statistical differences between the total distance travelled in the open field test by the different experimental groups (SYN120 tg mice treated with vehicle, MPH or I-*threo*) were assessed by using two-way ANOVA followed by Bonferroni’s multiple comparisons test (n = 3–4 animals for each group). Differences between the time to transverse the beam of vehicle, MPH or **I-*threo***-treated SYN120 tg mice in the balance beam test were assessed by using one-way ANOVA followed by Newman–Keuls multiple comparisons test (n = 3–4 animals for each group).

### 2.12. Prediction of Physicochemical Descriptors

For all the studied compounds, physicochemical descriptors relevant for drug-likeness, ADME parameters and pharmacokinetic properties were predicted using the SwissADME tool (www.swissadme.ch, accessed on 17 November 2021, Molecular Modelling Group—Swiss Institute of Bioinformatics, Lausanne, CH) [41,42]. Results were plotted as radar graphs, highlighting the ideal chemical space for oral bioavailability.

### 2.13. Protein–Protein Docking

The files containing the atomic coordinates for the considered proteins were obtained from the RCSB Protein Data Bank (PDB, www.rcsb.org, accessed on 17 November 2021). Protein–protein docking was performed using the HDOCK server (hdock.phys.hust.edu.cn, accessed on 17 November 2021, Lab of Biophysics and Molecular Modelling—Huazong University, Huazong, CN) [43,44,45,46,47]. Syn III (PDB ID: 2P0A) [48] was loaded as the receptor and α-synuclein (PDB ID: 2KKW) [49] as the ligand, setting the other parameters to default. Chain A for model 2P0A and chain A.1 for model 2KKW were isolated from the PDB file to perform the docking. Model 2KKW was selected in agreement with previous computational studies carried out by our group on α-synuclein [16]. Out of the 10 top-scoring models generated by the docking protocol, model 1 was considered for the subsequent studies. Chimera molecular viewer (UCSF, San Francisco, CA, USA) [50] was used for the visualization of the output model, for surface generation and for the analysis of residues.

### 2.14. Docking Studies

The blind docking experiment of the studied ligands toward the previously obtained model of αSyn /Syn III complex was performed using Autodock Vina (Molecular Graphics Laboratory, Department of Integrative Structural and Computational Biology, The Scripps Research Institute, La Jolla, CA, USA) [51]. The number of docking poses was set to 10, with other Vina parameters set as default. Grid parameters are reported in the following: X = −10.0695, Y = −5.8370, Z = −15.5476; grid size 61.6413, 69.3213, 70.8827 Å (X, Y, Z). Output data, such as calculated binding energy and interaction patterns, were analyzed and scored using Chimera molecular viewer (UCSF, San Francisco, CA, USA) [50], which was also used to produce the artworks. In parallel, putative binding site analysis was carried out using the DeepSite function of PlayMolecule (playmolecule.org/deepsite, Accelera, Middlesex, UK) [52].

### 2.15. MD Simulations

MD simulations were carried out using PlayMolecule (Accelera, Middlesex, UK) starting from the output models of docking experiments. More in detail, ligand was prepared by running the Parametrize function (playmolecule.org/parameterize), based on GAFF2 force field [53]. The αSyn/Syn III complex was prepared for the MD simulation using ProteinPrepare (playmolecule.org/proteinPrepare) and SystemBuilder (playmolecule.org/SystemBuilder) functions, setting pH = 7.4, AMBER force field and default experiment parameters [54]. Simulation runs of 25 ns in presence of water in a globular system were carried out using SimpleRun tool with default settings (playmolecule.org/SimpleRun). Plotting of RMSD values over time was performed using Excel 15.31 (Microsoft Corporation, Redmond, WA, USA).

## 3. Results and Discussion

### 3.1. FRET Studies Assessing the Ability of MPH Analogues to Influence the Interaction between αSyn and Syn III in SK-N-SH Cells Overexpressing RFP-Tagged αSyn and GFP-Tagged Syn III

First, we evaluated the ability of MPH analogues **I**–**IV** to modulate the interaction between αSyn and Syn III in a cell model of PD exhibiting αSyn and Syn III co-aggregates by acceptor photobleaching FRET as previously described [16]. This technique allows the assessment of protein–protein interaction through the measurement of FRET efficiency [33,34,35,36,37]. In particular, we recently reported that acceptor photobleaching FRET allows us to detect the increase in the functional αSyn/Syn III interaction in SK-N-SH cells overexpressing RFP-αSyn and GFP-Syn III and exhibiting co-aggregates of the tagged proteins [16]. The FRET efficiency, which relies on the proximity of the FRET couple, and thus to GFP-Syn III/RFP- αSyn interaction in our case, was measured as the recovery of the donor GFP-Syn III signal after the complete bleaching of the acceptor RFP-αSyn. For the FRET experiments basal measurement were performed on control cells cultured in conventional media, while the measurements for the evaluation of the ability of MPH, **I-*threo***, **I-*erythro***, **II-*threo*** and **IV-*threo*** to promote αSyn/Syn III interaction were performed at 15 min from the addition of 10 μM of each of the different compounds to the cell culture media as previously described [16].

First, we confirmed that MPH treatment significantly increased αSyn/Syn III interaction in vitro (Figure 2B). The MPH analogues **I-*threo*** and, to a lesser extent, **IV-*threo*** were found to significantly increase FRET efficiency and thus Syn III and αSyn binding (Figure 2B). Notably, compound **I-*threo*** exhibited a higher FRET efficiency when compared to MPH, supporting that it is more effective than MPH in inducing αSyn/Syn III interaction. This observation suggests that the substitution of the phenyl group with a *p*-tolyl increases the affinity for the pocket formed by the interaction between αSyn and Syn III. The **I-*erythro*** isomer, conversely, did not show any effect (Figure 2B).

We then used FLIM-based FRET analysis in GFP-Syn III and RFP-αSyn overexpressing SK-N-SH cells to confirm that compound **I-*threo*** has an improved capability to stimulate αSyn/Syn III interaction when compared to MPH. This method measures the lifetime of the excited state of the donor that, in our case, was GFP-Syn III. As the lifetime depends on the energy transfer between the donor and the acceptor, when FRET occurs, it is decreased. 

In the cells treated with MPH or **I-*threo****,* the donor lifetime was reduced, supporting the occurrence of the energy transfer between the GFP-RFP pair and thus the interaction between Syn III and αSyn (Figure 2C and Appendix A). These results confirm that the modifications introduced on the MPH structure leading to compound **I-*threo*** improve the ability to stimulate αSyn/Syn III complex. Remarkably, FLIM analysis also confirmed that the **I-*erythro*** stereoisomer did not increase αSyn/Syn III interaction (Figure 2C and Appendix A).

### 3.2. MPH Analogues Did Not Exhibit Toxic Effects at the Concentrations Stimulating the Functional αSyn/Syn III Interaction on Primary Mesencephalic Neurons

We then assessed the toxicity of the compounds on primary mesencephalic neurons. Cells were treated with increasing concentrations of the MPH analogues and then subjected to MTT assay to determine cell viability after 24 h. None of the assessed compounds, except for **III-*threo***, impacted cell viability at the concentration stimulating αSyn/Syn III interaction (10 μM). Compounds **I-*threo*** and **II-*threo*** significantly reduced cell viability only at 100 μM concentration, which is 10 times higher than the effective concentration tested in the FRET and FLIM experiments (Figure 2C and Figure 3). The only compound that exhibited a high rate of toxicity at the working concentration of 10 μM was **III-*threo***, which is unable to stimulate the functional αSyn/Syn III interaction but significantly reduced cell viability at concentrations as low as 0.1 μM (Figure 3).

### 3.3. Compound I-threo Reduced αSyn Aggregation in SK-N-SH Cells Overexpressing αSyn and in Primary Mesencephalic Neurons Exposed to GD

As compounds **I-*threo*** and **IV-*threo*** were found to efficiently modulate αSyn/Syn III without exerting toxic effects on the cells, we evaluated their effect on the aggregation of αSyn. First, we tested the compounds on SK-N-SH cells overexpressing human full length (fl) αSyn, an established in vitro model of PD, as three days of human αSyn overexpression is sufficient to induce the formation of aggregates [40].

Briefly, αSyn overexpressing SK-N-SH neuroblastoma cells were fixed 24 h after treatment with 10 μM **I-*threo*** or **IV-*threo*** and stained for αSyn. A negative control was performed by omitting the incubation with the primary antibody (Appendix A). By confocal microscopy, we observed that αSyn-overexpressing cells presented an αSyn punctuate staining, confirming the presence of αSyn aggregates as previously described (Figure 4A [16,40]).

After treatment with compounds **I-*threo*** or **IV-*threo***, the cells displayed a more diffused staining with small positive dots dispersed in the cell body (Figure 4A). The quantification of the size of αSyn-immunopositive particles confirmed that both **I*-threo*** and **IV-*threo*** were able to significantly reduce the dimension of αSyn immunopositive dots (Figure 4B), suggesting that treatment for 24 h with these two compounds was able to hinder the aggregation of αSyn. The effect of compound **I-*threo*** on αSyn aggregation was confirmed also in primary mesencephalic neurons. In particular, these primary cells were maintained in culture till DIV 7, in order to allow their maturation and were then exposed to GD, an insult that induces αSyn aggregation [3,55].

Cells were then treated with two different concentrations of compound **I-*threo*** or **IV-*threo*** to assess their ability to prevent αSyn aggregation. Since **I-*threo*** is known to have high binding affinity for DAT, lower concentrations of both the compounds (0.1 μM and 0.05 μM) were used to reduce off-target effects on neuronal cells (compared to the SK-N-SH cells treatments). We observed that 0.1 μM of compound **I-*threo*** was required to significantly decrease the size of αSyn-positive particles (Figure 5A,B) after 24 h of treatment. The treatment of GD-exposed cells with the same concentration of compound **IV-*threo*** induced only a slight and not significant decrease in αSyn-positive dots (Figure 5A,B), suggesting that the presence of the N-methylated amine reduces its efficacy in primary neuronal cells.

Collectively, this evidence further support that compound **I-*threo*** reduces αSyn aggregation in vitro, suggesting that by promoting the functional interaction between Syn III and αSyn, it decreases the ability of Syn III to stabilize αSyn aggregates. This effect is analogous to that observed following Syn III gene-silencing-based reduction [3,13].

Since the results on primary mesencephalic neurons supported that compound **I-*threo*** could significantly reduce αSyn aggregation at 0.05 μM concentration, we tested the effect of lower concentrations of this compound on αSyn/Syn III interaction and αSyn aggregation in SK-N-SH cells (Appendix A). We found that neither 0.05 nor 0.1 μM **I-*threo*** administration could improve αSyn/Syn III interaction as detected by FRET nor reduce αSyn aggregation. The different effects of **I-*threo*** on αSyn-overexpressing SK-N-SH cells and GD-exposed primary mesencephalic neurons can be ascribed to the fact that, in SK-N-SH cells, αSyn aggregation is related to the overexpression of the protein. This can have a stronger impact on the amount and size of the generated aggregates, which are thus predicted to be higher than in the GD-exposed primary mesencephalic neurons.

### 3.4. The Acute i.p. Administration of 10 mg/kg of Compound I-threo Enhanced the Locomotor Activity of 12-Month-Old αSyn tg Mice Significantly More Compared to MPH

As **I-*threo*** was found to be efficient in counteracting αSyn aggregation in vitro, the compound was also tested in vivo in 12-month-old SYN120 human αSyn tg mice expressing the human C-terminally truncated 1–120 form of αSyn in the nigrostriatal system on an αSyn null background. This is a well-established model of early PD that, at 12 months of age, exhibits marked αSyn aggregation, DAT redistribution and DA release reduction, as previously described [3,16,56]. 

We also recently showed that 12-month-old SYN120 tg mice are responsive to MPH acute treatment, exhibiting an increased locomotor response to the drug, that results Syn III-dependent. Indeed, this motility response is abolished by Syn III gene silencing but does not rely on DAT inhibition [16]. This is consistent with the fact that in the 12-month-old SYN120 tg mice, DAT is retained in αSyn intrasynaptic inclusions, as occurs in PD patients and other in vivo PD models [23,24,36]. For this reason, DA reuptake ability is compromised in the SYN120 tg mice, as supported by evidence showing that these animals do not respond to the acute administration of GBR-12935 [16] and also exhibit a very scarce increase in DA release following cocaine stimulation [13].

Therefore, we evaluated whether a single acute treatment with compound **I-*threo*** could improve the motility of SYN120 tg mice in a manner similar to MPH. By assessing the total distance travelled in an open field arena as previously described, we found that the acute administration of MPH 10 mg/kg i.p. induced a significant increase in the motility 10 min after drug injection (20.0 min, Figure 6A) in line with our previous observations [16]. Remarkably, we found that the locomotor response induced by the acute administration of 10 mg/kg of compound **I-*threo*** was significantly more pronounced than the one observed following MPH acute treatment starting from 7.5 min from the injection (22.5 min, Figure 6A). In addition, by using the beam walking test, we observed that compound **I-*threo*** was significantly more effective than MPH in reducing the time to traverse the beam (Figure 6B). These findings corroborate the results from FRET- and FLIM-based analyses, which showed that compound **I-*threo*** stimulates the functional interaction between αSyn and Syn III more efficiently than MPH.

### 3.5. Drug-Likeness

Considering the potential of the MPH analogues we selected and prepared, we decided to calculate their physicochemical descriptors, and the studied compounds can be defined as drug-like molecules. In fact, ideal values for pharmacokinetic parameters including lipophilicity, size, polarity, solubility, unsaturation, and flexibility were predicted. Overall, the compounds fall within the suitable physicochemical space for orally bioavailable compounds. More specifically, based on the computed physicochemical descriptors, the compounds are predicted to reach the central nervous system through passive blood–brain barrier permeation [41]. For reference, the same parameters were also computed for MPH, and comparable values were observed (Appendix A).

### 3.6. Generation of the αSyn/Syn III Complex Model

Protein–protein docking was enrolled to build a putative model of the complex formed by αSyn and Syn III using HDOCK. Interestingly, in 5 out of 10 generated complexes, partial co-localization of docked αSyn in a single binding site on the surface of Syn III was observed (all the models are reported in Appendix A). The best scoring model (ΔG = −242.96 kcal/mol, RMSD = 124.90 Å) was selected, studied more in detail, and used for subsequent investigations. The predicted complex is represented in Figure 7A, where Syn III is depicted in blue and αSyn in grey. In the same figure, the residues which are located at the interface between the two proteins have been labelled (Figure 7B). In detail, residues 10–34 and 59–77 of αSyn are involved in the binding with Syn III. Most interestingly, this second portion of the protein, together with residues 238–276 of Syn III, forms a pocket which may represent a novel putative binding region for ligands targeting the assembly constituted by these macromolecules. Consistently, DeepSite analysis, which was carried out in parallel on this complex, highlighted the presence of the main potential ligand interaction site at the interface between the two proteins in this same region (Appendix A).

### 3.7. Docking Studies and Molecular Dynamics Simulations

A blind docking approach was adopted to confirm the interaction site and explore more in detail the binding mode of the studied compounds towards the αSyn/Syn III assembly. The compounds were observed to form three clusters upon binding to the complex. More specifically, this set of experiments highlighted that most of the tested molecules interact with the pocket formed at the interface between the two proteins (Figure 8A). Nevertheless, only one compound, **II-*threo***, interacts with a different portion of the macromolecule and interferes with two smaller binding sites on Syn III, which were also previously detected by the DeepSite analysis as secondary pockets (Appendix A). Even if with different orientations, the compounds targeting the main pocket bind the same pool of residues. Interestingly, observations from blind docking agree with the results of DeepSite analysis, which highlighted the presence of a wide ligand interaction site at the interface between proteins (Appendix A). Hydrophobic interactions are involved in the recognition process (Figure 8B). The binding motif of MPH and **I-*threo*** is depicted in a more detailed fashion in Appendix A: Val63, Val66 and Val70 are the residues in closest proximity to both compounds, according to the predicted models.

The binding energies calculated for all the tested derivatives fall within a rather narrow range (Appendix A). MPH and **I-*threo*** showed identical scores, the lowest among the screened compounds in absolute value. Compound **IV-*threo***, which exhibited almost the same capability to stimulate the functional αSyn/Syn III interaction of MPH, also exhibited similar binding energy. The calculated binding energy values for **I*-erythro***, **II-*threo*** and **III*-threo***, which were unable to stimulate the functional αSyn/Syn III interaction, were more promising than that computed for MPH, **I-*threo*** and **IV-*threo***. These findings suggested that more advanced investigations, such as MD simulations, may offer a better perspective to understand the binding of the active compounds to the αSyn/Syn III complex.

Therefore, 25 ns MD simulations were performed to provide a preliminary insight on the stability over time of the complex between MPH or **I-*threo*** and the αSyn/Syn III assembly. Results showed that in both cases, the αSyn/Syn III complex was stable over simulation time, with very limited variations of root-mean-square deviation (RMSD). On the other hand, MD simulations predicted that while **I-*threo*** could be retained within the binding site with small fluctuations, MPH did not reach stabilization within the simulation timeframe, showing high RMSD values that are not consistent with a structured ligand-target binding (Appendix A). These observations are fully in line with the results from FRET and FLIM experiments supporting that **I-*threo*** have an improved capability to stimulate the functional αSyn/Syn III interaction when compared to MPH.

Considering the potential of the MPH analogues we selected and prepared, we decided to calculate their physicochemical descriptors, and the studied compounds can be defined as “drug-like”. The acute i.p. administration of 10 mg/kg of compound I-threo enhanced the locomotor activity of 12-month-old αSyn tg mice significantly more compared to MPH.

### 3.8. Study of the DAT Binding Activity of Compound I-threo

Previous reports assessed the DAT-inhibitory action of compound **I-*threo***, showing that it is almost comparable to that of MPH [26]. However, to the best of our knowledge, DAT affinity (K_i_) of compound **I-*threo*** has never been determined. In light of the above-described results, we thus evaluated the DAT K_i_ of **I-*threo*** and of MPH as reference control (Appendix A). Our results showed that **I-*threo*** exhibited an improved DAT binding ability when compared to MPH.

A summary of previously reported K_i_ and/or IC_50_ data of MPH, **I-*threo***, **I-*erythro***, **II-*threo***, **III-*threo*** and **IV-*threo*** is reported in Appendix A.

## 4. Conclusions

The results presented above support that among the selected MPH derivatives, compound **I-*threo*** exhibited an improved ability to stimulate the functional αSyn/Syn III interaction. These results were also corroborated by our in silico MD studies on the generated complex composed of α-helical αSyn and Syn III. Indeed, although docking studies showed that MPH and **I-*threo*** exhibited a comparable binding energy, MD predictions supported that the latter could form a stable interaction with the αSyn and Syn III complex, while MPH did not reach stabilization in the simulation timeframe.

Interestingly, compound **I-*threo***, which exhibited an improved ability to stimulate the functional interaction between αSyn and Syn III when compared to MPH, was also more effective in reducing αSyn aggregation.

In line with our previous observations, supporting that the in vivo reduction in Syn III achieved by RNA interference in SYN120 tg mice enables the disruption of αSyn aggregates even when these are already formed and animals exhibit a frank synaptic dysfunction [13], our results support that by promoting the functional interaction between αSyn and Syn III, MPH and the derived compound **I-*threo*** may sequester Syn III from the pathological deposits, thus significantly reducing αSyn aggregation in in vitro models.

More strikingly, we observed that the motor recovery effect observed upon the acute administration of compound **I*-threo*** in the SYN120 tg mouse line was higher than that exerted by MPH administration. We previously demonstrated that the motor recovery effect of MPH in this mouse line cannot be ascribed to DAT binding, as striatal DAT is clumped within αSyn aggregates [23] and GBR−12935 cannot block MPH-induced motor recovery, which is instead abolished by Syn III gene silencing [16]. Although when compared to MPH compound **I-*threo*** exhibits an improved DAT affinity, as supported by our data, but an almost comparable DAT inhibitory action, as previously reported [26], the inability of GBR−12935 to hamper MPH-mediated motor recovery in the SYN120 tg mice indicates that the effect of compound **I-*threo*** cannot be related to DAT inhibition.

Previous studies assessing the DAT inhibitory activity of a series of MPH derivatives, encompassing compounds **I-*threo***, **I-*erythro***, **II-*threo***, **III-*threo*** and **IV-*threo***, have shown that the *erythro* diasteroisomers of MPH derivatives, analogously to MPH-*erythro*, do not block the DAT [26]. Conversely, some of the *threo* analogues, such as **II-*threo*** showed a very strong DAT binding affinity and inhibitory action [26,27,28,57]. This notwithstanding, we observed that **II-*threo*** was unable to stimulate the functional interaction between Syn III and αSyn.

Interestingly, even compounds **III-*threo*** and **IV-*threo***, which, from previous studies, exhibited very different DAT IC50 (10 times higher and almost comparable to MPH for **III-*threo* [26]** and **IV-*threo* [29]**, respectively), were both unable to stimulate Syn III/αSyn interaction.

The different behavior of MPH and its analogues towards DAT inhibition and Syn III/αSyn interaction, especially for compounds showing hindered aromatic moieties, suggests that the binding sites for these compounds on the targets are structurally different.

Overall, these results support the hypothesis that MPH and MPH analogues, having an αSyn/Syn III complex stabilizing ability, may sequester Syn III from the pathological αSyn aggregates, thus enabling their reduction analogously to Syn III gene silencing. More strikingly, through this mechanism, these compounds restore the functional αSyn/Syn III interaction, which is pivotal for ensuring DA release [3,13,16].

Collectively, our findings highlight **I-*threo*** as a suitable lead compound for counteracting motor dysfunction and reducing αSyn pathology in PD and could be the starting point for further structural modifications aimed at avoiding unwanted activities as MAT binding, to design novel drug candidates exerting a disease-modifying effect on αSyn aggregation in PD. These findings have significant implications in the field of PD and other synucleinopathies, but may even open insightful therapeutic perspectives in the field of ADHD, which can be associated with Syn III and αSyn polymorphisms [14,58,59] and is conventionally treated with MPH.

## 5. Patents

WO2022/029151 A1: Structural analogues of Methylphenidate as Parkinson’s disease-modifying agents.

## Data Availability

The data presented in this study are available on request from the corresponding author.

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
