# Peer review of "Methylphenidate Analogues as a New Class of Potential Disease-Modifying Agents for Parkinson’s Disease: Evidence from Cell Models and Alpha-Synuclein Transgenic Mice"

_pharmaceutics, 2022, doi:10.3390/pharmaceutics14081595_

Round 1

Reviewer 1 Report

This study is a comprehensive and well designed. The author identified and thoroughly tested a new lead compound that show clear beneficial effect to modulate alpha-Syn/SynIII interaction and subsequently reduce alpha-Syn aggregates. The studied fully validate the function of I-thero both in vitro and in vivo. I only have some minor suggesitons:

1. It will be of important to address the toxicity of the I-thero compound both in vitro and in vivo. 

2. The results from Figure3 and Figure4 seams controversial regarding to the function of I-thero and IV-thero, a higher dosage of IV-thero need to be checked regarding to cell viability. While it will also be interesting to explore whether lower dosage of II-thero or III-thero might work for reduce aggregations.

3. The author need to explain or discuss more about the reason/mechanisms that bring different results between SK-N-SH cells and primary neurons regarding to the similar rescue effects in SK-N-SH cells while different rescue effects in primary neurons by using same dosage of I-thero and IV-thero.

4. At least one extra behavior experiment need to be added to support the in vivo observation by I-thero administration in mice.

Reviewer 2 Report

The study by Casiraghi is well written, concise and of broad interest to the field. The experimental design is well controlled and detailed, and the conclusions are sound. However, the title should be modified to reflect the study. By this I mean the title should reflect that this work was completed using in vitro and in vivo Tg mouse models. 

-Have the authors pre-treated their SK-N-SH cells over-expressing SNCA with lysosomal and proteosomal inhibitors? I ask this question because the data presented suggest that there is a reduction in aggregate size upon treatment with the stated compounds, not a reduction in total protein level. 

-In a similar vein, have the authors looked at levels of lysosomal proteins associated with the degradation of alpha-synuclein, such as Cathepsin D, GBA etc. Is it possible the proteins may change upon treatment with the listed compounds?

-Have the authors investigated the general endosomal morphology following treatment with the stated compounds? this will help determine if  the changes observed reflect a more global loss of endosomal homeostasis or a specific phenotype observed that can be attributed to compound use.

Round 2

Reviewer 1 Report

The revised version addressed most of my concerns, need some additional spell and grammar checking before publish.